# Development of a Digital Healthcare Management System for Lower-Extremity Amputees: A Pilot Study

**DOI:** 10.3390/healthcare11010106

**Published:** 2022-12-29

**Authors:** Jin Hong Kim, Yu Ri Kim, Mi Hyang Han, Ji Young Lee, Ji Sung Kim, Yong Cheol Kang, Seong Jun Yoon, Yunhee Chang, Gangpyo Lee, Nam Soon Cho

**Affiliations:** 1Rehabilitation Medical Research Center, Incheon Hospital, Korea Worker’s Compensation & Welfare Service, Incheon 21417, Republic of Korea; 2Department of Physical and Rehabilitation Medicine, Incheon Hospital, Korea Worker’s Compensation & Welfare Service, Incheon 21417, Republic of Korea; 3Rehabilitation Engineering Research Institute, Korea Worker’s Compensation & Welfare Service, Incheon 21417, Republic of Korea

**Keywords:** lower-extremity amputees, physical-strength training, information and communication technology

## Abstract

The research, which was designed as a “pre- and post-single group” study, included patients with lower-limb amputation and aimed to evaluate the effectiveness of self-directed physical-strength training and cardiovascular exercise using a novel digital healthcare management service three times a week for 12 weeks. Muscle strength, thigh circumference, lipid profile and glycated hemoglobin levels, pulmonary function, quality of life, and physical activity level were evaluated before and after the intervention, while satisfaction was measured after the study. Among the 14 included patients, the proportion of adherence to the physical-strength training and physical-strengthening activity were 85.2% and 75.8%, respectively. The level of satisfaction with the digital healthcare management system was high. Significant changes were observed in the muscle-strength tests (dominant grip power and muscle strength of knee flexion and extension of the intact side), thigh circumference, and glycated hemoglobin levels. Further, the quality-of-life score showed improvement, although without significant differences. Individualized exercise management using the novel digital healthcare management system for lower-limb amputees could induce interest in self-care and promote physical activity and healthy behavior. Through this effect, we can expect a reduction in the incidence of cardiovascular diseases, diabetes mellitus, dyslipidemia, and severe injuries from falling.

## 1. Introduction

Amputation is a procedure that involves the removal of a body part. Acquired amputation results from an operation that aims to remove a body part following trauma or disease [1]. Further, a physical disability caused by the loss of limb function due to amputation of the upper or lower extremities is known as an amputation disability. Accordingly, lower-extremity amputees (LEAs) have reduced gait function, less energy efficiency during walking, and difficulties managing prosthetic devices [2]. Moreover, they often experience pain, falling, and chronic wounds due to prosthetics; hence, they experience restrictions on modifying their lifestyle and participation in regular exercise [3]. LEAs often suffer from musculoskeletal diseases, are less physically active, and are vulnerable to cardiovascular diseases [4]. Therefore, amputee rehabilitation should include a long-term health management strategy after acute-phase prosthetic rehabilitation [5]. Considering the clinical characteristics and the importance of the general health and exercise management of LEAs, a comprehensive approach for managing appropriate lifelong health is essential.

Digital health encompasses various platforms and systems that apply technological solutions to enhance healthcare delivery. Digital therapeutics (DTx) is one such category of digital health solutions that provides evidence-based, software-driven therapeutic interventions for the prevention and management of a medical disorder or disease [6]. Recently, a medical service model through a digital healthcare system has gained applications in the clinical field [7,8,9,10]. 

A web-based coaching program for clinicians was developed to manage the amputees in general outpatient clinics. Here, we designed a health-management program for LEAs that included self-directed exercise and health management with regular medical clinic supervision [11]. To improve this program’s adherence and effectiveness, we applied a digital healthcare system that contained mobile applications, wearable devices for amputees, and a health check-up booth in the hospital. This study aimed to evaluate the effect of our novel digital healthcare management system on LEAs.

## 2. Materials and Methods

### 2.1. Digital Healthcare Management System for Lower-Limb Amputees

A model of healthcare for patients with lower-limb amputation (LLA) was developed based on that implemented in a previous study [12]. First, Internet of Things (IoT)-based sensors for home and hospital use were connected to monitor the patient’s health status, and self-directed exercise- and health-promotion service platforms were developed. The IoT-based sensors for home use included a sphygmomanometer, a blood glucose monitor, and a device to monitor physical activity (wearable devices). For hospital use, a kiosk for a personal identity verification process that included a health check-up booth, a body-composition scale for the disabled, a stress tracker, a handgrip dynamometer, and a sphygmomanometer for medical use were included. Second, a program for managers to monitor patients and prescribe exercise, a mobile application (app), and a web program for patients to perform self-directed healthcare were developed. Third, a data-storage server was constructed for personal health information and security technology. The patient general health data (PGHD) measured via the health check-up booth (once a month) for hospital use, the IoT sensor devices for home use (daily), and the wearable devices for patients (daily) were stored in the mobile app and on the web server. The family doctor monitored the PGHD stored on the web for managers and prescribed an exercise program that was appropriate according to the patient’s health level. The mobile app for patients was used to check and run the prescribed exercise program. Accordingly, the history of exercise would be recorded through the mobile app and wearable device, and the family doctor would periodically monitor the recorded history of exercise and provide feedback to the patients (Figure 1).

### 2.2. Participants

The research, which was designed as a “pre- and post-single group” study, included 15 patients who visited the outpatient rehabilitation center of the Korea Workers’ Compensation & Welfare Service’s Incheon Hospital. The inclusion criteria were as follows: patients without cognitive impairment, those who had undergone a unilateral LLA (a person with a below-ankle amputation was excluded), those with more than 1 year after the onset of disease, and those with a level 2 or above K-level. The exclusion criteria were as follows: patients with LLA who had history of stroke or traumatic brain injury, those who had surgery within six months due to abnormalities of the nervous and musculoskeletal systems, those who had never used a smartphone, and those who had difficulty taking part in the study at the discretion of the investigator. This study was approved by the Institutional Review Board of the Korea Workers’ Compensation & Welfare Service’s Incheon Hospital (No. KCIRB-2020-0003). The research protocol was registered with the Korean clinical trial registry (https://cris.nih.go.kr, accessed on 28 May 2020; CRIS identifier: KCT0005183).

### 2.3. Intervention 

The digital healthcare management system was used to perform home self-directed physical-strength training and cardiovascular exercise three times a week for 12 weeks. For the physical-strength training, video content for web-based three-dimensional animation exercise was developed by the study team, which comprised doctors and physical therapists for patients with LLA (Figure 2).

The exercise was composed of five steps in the order of warm-up, stretching, muscle strengthening, balance and stability exercises, and cool-down (Table 1; the videos are available at https://www.youtube.com/channel/UCHO5if0Rlz5zrG_rb_mvCAQ/videos (accessed on 28 May 2020).

Patients watched a series of exercise videos and followed the movements using the mobile app. A detailed description of the exercise and precautions were uploaded to YouTube, and the mobile app provided a shortcut tab to YouTube. Depending on the type of disability, the amputation was classified into above- or below-knee amputation, and there were three difficulty levels. A modification was made after the patients visited the outpatient rehabilitation center and consulted the family doctor once every four weeks. For the cardiovascular exercise, the patients took a 30 min walk each day with an intensity equivalent to 60–80% of each patient’s maximum heart rate; the walks lasted 20 min during the first week and 30 min during weeks 2–12. Accordingly, the age-predicted maximum heart rate was calculated by subtracting a patient’s age from 220 [13]. 

### 2.4. Measurement Methods and Items

The patients’ clinical data and self-reported measurements were obtained before the study and after 12 weeks of exercise. A satisfaction survey on the digital healthcare management system was conducted one week after the end of the study. Since adherence was a core element to help us understand the clinical efficacy, measurement of the adherence was calculated by inputting the data recorded into the calculation formula on the manager servers.

### 2.5. Clinical Data

#### 2.5.1. Muscle Strength

The lower-limb muscle strength was assessed using a Micro-FET3 handheld muscle-testing dynamometer (Hoggan Scientific, Salt Lake City, UT, USA), which is an isometric digital muscle-measurement tool. The muscle strength of knee flexion and extension in the intact knee was measured while the participants lay on their face. The grip strength of the dominant hand was measured using a Jamar digital dynamometer (JLW instruments, Chicago, IL, USA). The grip strength was measured while a participant was sitting on a chair without armrests. The patient was asked to sit with their shoulder adducted, elbow flexed to 90 degrees, and forearm and wrist neutral. The participant was asked not to move while measuring the grip strength [14]. The test was repeated twice by the same tester to reduce measurement errors and calculate an average.

#### 2.5.2. Thigh Circumference Test

Thigh circumference of the intact side was measured using the method proposed by Ronald McRae [15]. After the patients (who were wearing shorts) were asked to naturally flex their leg, the point on the thigh 18 cm from the knee joint line in the direction of the body was marked. The diameter of the marked point was measured using a tape measure.

#### 2.5.3. Lipid Profile 

All patients were instructed to fast for at least 8 h on the test day. After resting for approximately 30 min upon arrival at the laboratory, 10 mL of blood was drawn from a vein in the forearm. The serum was separated from the blood via centrifugation. Without the process of freezing, the total cholesterol (TC), triglyceride (TG), high-density lipoprotein (HDL-C), and low-density lipoprotein cholesterol (LDL-C) levels were analyzed at a clinical center of Incheon hospital.

#### 2.5.4. Glycated Hemoglobin Examination

The glycated hemoglobin (HbA1c) values, which were obtained from the medical records documented with the values, were analyzed using high-performance liquid chromatography when testing for dyslipidemia. In this regard, an HbA1c level of ≤6.5% was considered to be a controlled blood glucose level [16].

#### 2.5.5. Pulmonary Functional Test 

A pulmonary function test was conducted using a spirometer (Vyntus spiro, Vyaire Medical, Chicago, IL, USA). A mouthpiece with a nose clip was used to ensure that no air escaped from the nose when breathing through the mouthpiece during a spirometry test. The participants were fully educated so that no air escaped through the nose [17]. The forced vital capacity (FVC) and forced expiratory volume in 1 s (FEV1) were measured while the participants were in a sitting position. With the start sign of the tester, the participant made a usual inspiration and expiration two to three times and then inspired the maximum amount of air quickly and expired as much as possible. The measurements were conducted by repeating this three times [6].

### 2.6. Self-Reported Measures

#### 2.6.1. Quality of Life

The European Quality of Life 5 Dimensions 3 Level Version (EQ-5D-3L) was used to measure the health-related quality of life. The EQ-5D-3L includes five dimensions (mobility, self-care, usual activities, pain/discomfort, and anxiety/depression) with three domains per dimension. Since it is easy and simple for responders to complete in a short time, it has been widely used in various studies on the general public and patients [18]. In this study, the Korean version was used [19]. 

#### 2.6.2. Amount of Physical Activity

The International Physical Activity Questionnaire (IPAQ) with the self-reported short form was used. The questionnaire included questions about physical activity at the vigorous/moderate/low-intensity level performed for ≥10 min 1 week ago, and the time and number were subjectively answered. Based on the questionnaire data, we transformed the physical activity into metabolic equivalent task-minute (MET-min) scores in accordance with the IPAQ-SF sum score [20]. The Korean version was used [21]. 

#### 2.6.3. Satisfaction Survey on the Digital Healthcare Management System

The satisfaction survey on the digital healthcare management system that we designed consisted of nine items with a total of 50 points. Item numbers 1 to 8 consisted of questions about healthcare, device utilization, motivation, and future utilization using a five-point Likert scale. The Likert scale had the following options: strongly disagree (one point), disagree (two points), neutral (three points), agree (four points), and strongly agree (five points). Item number 9 measured satisfaction with the overall service on a scale of 1–10. A higher sum of the scores of item numbers 1 to 9 meant that the level of satisfaction was high.

#### 2.6.4. Adherence

The adherence to the physical-strength training was calculated based on the video playback time recorded in the app. The calculation formula was as follows: 60 min/day × 3 times/week × 12 weeks = 2160 min; playback time/2160 min × 100 = adherence (%). 

The adherence to the cardiovascular exercise was calculated based on the physical activity time of the heart rate interval, which was set for individual patients. The calculation formula was as follows: 20 min/day × 3 times/week = 60 min, 30 min/day × 3 times/week × 11 weeks = 990 min; physical activity time/1050 (60 + 990) min × 100 = adherence (%).

#### 2.6.5. Data and Statistical Analysis

The descriptive statistics were reported as proportions or means ± standard deviations. Differences between pre- and post-intervention values of the continuous variables were analyzed using the paired t-test or Wilcoxon signed-rank test. Two-sided *p*-values < 0.05 were considered statistically significant. The analysis was performed using IBM SPSS Statistics for Windows, Version 25.0 (IBM Corp., Armonk, NY, USA). A minimum sample size of 15 participants was calculated using a power-calculation tool (G* Power 3.1.9.3 software; Heinrich-Heine University, Düsseldorf, Germany) with the power, alpha, and effective size set at 0.08, 0.05, and 0.82, respectively

## 3. Results

One patient withdrew from participation in the study, so 14 patients were included. In terms of the general characteristics, the mean age was 46.1 years and all 14 were men, including 6 (42.9%) with an above-knee amputation and 8 (57.1%) with a below-knee amputation. The mean BMI was 29.1 (kg/m^2^) (Table 2). The proportions of adherence to the physical-strength training and cardiovascular exercise were 85.2% and 75.8%, respectively. A satisfaction survey on the digital healthcare management system received 41.3/50 points (82%), which indicated a relatively high level of satisfaction.

### 3.1. Clinical Data

The muscle strength of the lower limb in patients with amputation was assessed and revealed values of 162.1 ± 60.44 N before the intervention and 189.9 ± 41.4 N after the intervention. The muscle strength of knee flexion before and after the intervention were 266.4 ± 35.9 and 309.1 ± 27.8 N, respectively. A statistically significant improvement in muscle strength was observed.

The test of grip strength revealed that the grip strength was 45.9 ± 87.1 N before the intervention, which then increased to 47.6 ± 7.8 N after the intervention. A statistically significant improvement in grip strength was observed (Table 3).

The thigh circumference was 51.9 ± 4.0 cm and 53.3 ± 4.2 cm before and after the intervention, respectively, which showed a statistically significant change (Table 3).

The lipid profile revealed that the TC levels before and after the intervention decreased from 205.1 ± 24.9 to 202.0 ± 27.3 mg/dL, whereas the HDL levels before and after the intervention increased from 48.9 ± 11.8 to 51.9 ± 15.2 mg/dL. Further, the LDL levels before and after the intervention were 127.5 ± 19.3 and 122.2 ± 20.4 mg/dL, respectively. The TG levels before and after the intervention were 209.9 ± 125.6 and 186.1 ± 92.2 mg/dL, respectively. Accordingly, no statistically significant changes were observed.

The HbA1C level was 5.8% before the intervention and 5.5% after the intervention, which showed a statistically significant change (Table 3).

The pulmonary functional tests performed in patients with amputation revealed that the FVCs before and after the intervention were 4.2 ± 0.7 and 4.2 ± 0.9 L, respectively. The FEV1s before and after the intervention were 3.4 ± 0.6 and 3.4 ± 0.6 L, respectively. No statistically significant changes were observed (Table 3).

### 3.2. Self-Reported Measures

The scores regarding the quality of life measured in the patients with amputation were 16.3 ± 3.2 and 16.9 ± 2.1 points before and after the intervention, respectively, which showed no statistically significant changes.

The scores regarding changes in physical activity of disabled patients with amputation were 4806 ± 3875.3 and 4855.1 ± 3849.9 (MET-min/week) before and after the intervention, respectively. No statistically significant changes were observed (Table 4).

## 4. Discussion

To our knowledge, this was the first study to evaluate the effectiveness and safety of self-directed physical-strength training and cardiovascular exercise using a digital healthcare management system developed for patients with LLA. This study showed a statistically significant improvement in the grip strength, lower-limb muscle strength, thigh circumference, and HbA1c level in patients with LLA.

Skeletal muscle plays an important role in the metabolism and activities of daily living [22]. Reduced skeletal muscle mass and function increase the risks of various chronic diseases, including insulin resistance, hyperglycemia, and atherosclerosis; they also induce disability, falls, and osteoporosis [23]. The measurements of grip strength and lower-limb muscle strength were used as a representative method that measured skeletal muscle in a previous study [24]. The lower-limb muscle and grip strengths were correlated with each other [25]. Therefore, increasing or maintaining the grip and the lower-limb muscle strength and mass plays an important role in a patient’s overall health. This study measured the muscle strength of knee flexion and extension of the intact limb, thigh circumference, and grip strength, all of which increased in a statistically significant manner. This meant that the physical-strength training that used the digital healthcare management system increased the muscle strength and mass of the skeletal muscle. A previous study of a 12-week personalized exercise program that consisted of stretching, muscle-strength training, balance training, and cardiovascular exercise in patients with LLA reported that the gait function and lower-limb muscle strength improved [26]. A study by Lee et al. that evaluated a web-based exercise program in older adults reported that the overall muscle strength improved, thereby improving gait and balance and preventing falls [27]. The significant effects of personalized and web-based exercise programs were similar to the results of this study, which used a system developed from a mixture of those two programs. In addition, the grip strength, lower-limb muscle strength, and thigh circumference increased in this study due to the lower-limb muscle-strength training along with the functional exercise used while considering the characteristics of the patients.

Resistance training improves the muscle mass and strength, physical function, mental health, bone density, insulin sensitivity, blood pressure, lipid profile, and cardiovascular health in adults. In addition, cardiovascular exercise increases insulin sensitivity, mitochondrial density, vascular compliance, vasoreactivity, pulmonary function, immune function, and cardiac output [28,29,30,31]. Furthermore, regular exercise reduces the resting heart rate, increases resistance to myocardial dysrhythmias, reduces atherosclerosis by improving blood lipids, and prevents or delays the onset of type 2 diabetes. Since serum lipid variables affect the metabolism of healthy individuals and patients in various ways, it is very important to keep serum lipid levels in healthy ranges [32]. Exercising reduces the levels of TC, TG, and LDL-C and increases the level of HDL-C [33,34]. Previous studies reported that a combination of cardiovascular exercise and resistance training was more effective than performing these alone and that performing moderate- to low-intensity exercise showed positive results on certain serum lipid variables compared to high-intensity exercise [35,36]. However, another study reported that no changes in serum lipid variables were observed after cardiovascular exercise and resistance training [37]. These conflicting study results may have been due to the differences in the experimental environments. This study did not show statistically significant changes in the serum lipid profile, which might have been due to the lower level of adherence to the cardiovascular exercise than the level of adherence to the physical-strength training. In contrast, the Hb1Ac test showed a statistically significant change. A study by Connelly et al. reported that for adults with type 2 diabetes, IoT-based physical activity that promoted intervention would be more effective than the general treatment [38]. Using devices such as wearable activity trackers and pedometers can change behaviors by monitoring physical activities [39]. In this study, behavioral changes may have resulted from the continuous monitoring of activity levels and blood glucose on the web using IoT-based wearable devices and blood glucose monitors.

The lungs acquire the oxygen required for exercising and release carbon dioxide from the body [40]. Since the body uses more energy during exercise, the oxygen demand increases. The lungs also play an important role in motor ability [40]. The FVC and FEV are physiological elements that signify the ability of pulmonary function [41]. In this study, no statistically significant changes in the FVC and FEV1 were observed. In a previous study that assessed the effect of trampoline training as a cardiovascular exercise for 8 weeks in adolescent participants, no significant change in the FEV1 was observed [42]. In a study that used an ICT-based healthcare service in patients with a spinal cord injury, an improvement in the pulmonary function was observed, although it was not statistically significant [43]. The results of this study were similar to those of previous studies, which might have been due to the absence of respiratory muscle training, which is a key factor in improving pulmonary function and in short-term interventions [43]. The World Health Organization defines the quality of life as an individual perception of a person’s life position in the context of culture and value system, as well as the tasks, expectations, and standards set by environmental conditions [44]. The quality of life may vary depending on multiple factors and can indicate various aspects of human life. A study by Grzebień et al. reported that among various factors that determined the quality of life of a patient after amputating their lower limb, appropriate treatment and rehabilitation should be the first priority [45] and that an increase in physical activities was a factor that influenced the quality of life. In this study, changes in the quality of life and activity levels were observed, although they were not statistically significant. In a recent study, the quality of life, physical activity, and opportunity for social engagement in Korean adults during the COVID-19 pandemic were lower compared to those during the pre-COVID-19 period [46]. This may have been due to the fact that more individuals spent a lot of time indoors by working at home and refraining from outdoor activities due to the spread and prolongation of COVID-19 while the study was conducted.

An average of 25% of patients do not comply with prevention and disease management activities (e.g., exercise, diet, medication, and screening) in the disease state. Adherence can drop to 50% or below in some medical conditions and environments [47]. To increase adherence, this study used a digital healthcare service that allowed for two-way communication between the doctor and patient and monitoring of the patients’ adherence to the exercise programs [48]. The results showed a relatively high level of non-adherence of 19.5%, which was lower than the mean level (25%) of non-adherence reported in previous studies [47]. The score for satisfaction in the satisfaction survey was high. In particular, the items that indicated that the device was helpful in health management and motivation showed high scores. The overall determination of medical outcomes and answers to the self-subjective survey questions showed that self-directed exercise using the digital healthcare management system could be considered a healthcare model for patients with amputation.

This study had some limitations. This study could not be generalized to all patients with LLA because we only included patients who met the inclusion criteria. In addition, we did not follow up regarding how long the effects lasted after the intervention. Since this was a pilot study, the sample size was small. There was a lack of information to verify the clinical efficacy because the study had no control group. Moreover, whether the patients’ physical-strength training and cardiovascular exercise performance rates were accurately measured could not be verified. Randomized controlled trials should be conducted by supplementing the limitations mentioned above and expanding the sample size to generalize the study results in the future. Furthermore, a strategic method should be established to reduce the study participants’ non-adherence to exercise programs.

## 5. Conclusions

Individualized exercise management through the digital healthcare management system for lower-limb amputees could induce interest in self-care and promote physical activity and healthy behavior. Through this effect, we can expect a reduction in the incidence of cardiovascular diseases, diabetes mellitus, dyslipidemia, and severe injuries from falling. However, further research with mid- to long-term monitoring and efforts to increase exercise performance are needed to confirm our expectations. 

## Figures and Tables

**Figure 1 healthcare-11-00106-f001:**
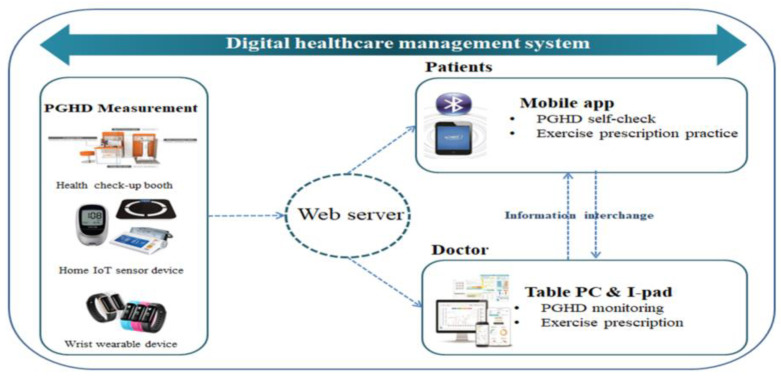
Digital healthcare management system.

**Figure 2 healthcare-11-00106-f002:**
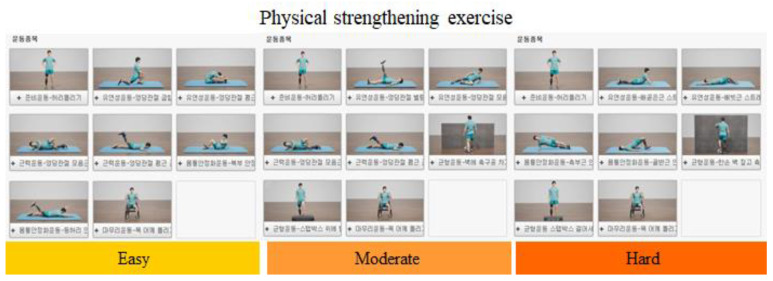
Three-dimensional animated exercise video content for lower-extremity amputees, including above-knee amputees.

**Table 1 healthcare-11-00106-t001:** Physical-strengthening exercises for lower-limb amputees.

No.	Type	Grade	Exercise Sequence
1	AK	Easy	1. Twisting of waist, 2. Hip flexor stretching, 3. Hip extensor stretching, 4. Hip adductor strengthening exercise, 5. Hip abductor strengthening exercise, 6. Sit-up, 7. Superman back extension, 8. Twisting of neck and shoulders.
2	AK	Moderate	1.Twisting of waist, 2. Hip abductor stretching, 3. Hip adductor stretching, 4. Hip extensor strengthening exercise, 5. Knee extensor strengthening exercise, 6. Kicking and rolling a ball, 7. Box step up and down, 8. Twisting of neck and shoulders.
3	AK	Hard	1. Twisting of waist, 2. Rectus abdominis stretching, 3. Internal/external oblique muscle stretching, 4. Side plank, 5. Bridge exercise, 6. Balance training with a ball, 7. Box step up and down, 8. Twisting of neck and shoulders.
4	BK	Easy	1. Twisting of waist, 2. Hip flexor stretching, 3. Hip extensor stretching, 4. Hip adductor strengthening exercise, 5. Hip extensor strengthening exercise, 6. Sit-up, 7. Superman back extension, 8. Twisting of neck and shoulders.
5	BK	Moderate	1. Twisting of waist, 2. Hip abductor stretching, 3. Hip adductor stretching, 4. Hip extensor strengthening, 5. Knee extensor strengthening, 6. Kicking and rolling a ball, 7. Box step up and down, 8. Twisting of neck and shoulders.
6	BK	Hard	1. Twisting of waist, 2. Knee flexor stretching, 3. Knee extensor stretching, 4. Side plank, 5. Bridge exercise, 6. Balance training with a ball, 7. Box jump up and down, 8. Twisting of neck and shoulders.

1. Turnaround of waist (10 reps on each side); 2,3. Stretching (five sets of 20 s each); 4,5. Muscle strengthening (five times of 20 reps, weight increase according to grade); 6,7. Balance and stability (five times of 20 reps); 8. Twisting of neck and shoulders (10 reps on each side). AK, above the knee; BK, below the knee.

**Table 2 healthcare-11-00106-t002:** General characteristics of the participants (n = 14).

Baseline Variable	M ± SD or Frequency (%)
Age (years)	46.1 ± 10.3
Sex (female/male)	0/14
Weight (kg)	73.4 ± 14.4
Height (cm)	170.1 ± 3.6
BMI (kg/m^2^)	29.1 ± 9.3
Level of injury (%)	AK: 6 (42.9)/BK: 8 (57.1)
K-level (%)	Level 2: 10 (71.4)/Level 3: 4 (28.6)
Onset duration (days)	676.5 ± 253.5

K-level, lower-limb extremity prosthesis Medicare Functional Classification level; M, mean; SD, standard deviation.

**Table 3 healthcare-11-00106-t003:** Changes in clinical data (n = 14).

Measures	Baseline	Post-Intervention	*t*-Value	*p*-Value
Muscle strength test				
Knee flexor (*N)*	162.1 ± 60.4	189.9 ± 41.4	3.15	0.00 *
Knee extensor (*N)*	266.4 ± 35.9	309.1 ± 27.8	5.20	0.00 *
Grip power (*N)*	45.9 ± 87.1	47.6 ± 7.8	3.11	0.00 *
Thigh circumference (cm)	45.9 ± 87.1	47.6 ± 7.8	3.11	0.00 *
Lipid test				
Total cholesterol (mg/dL)	205.1 ± 24.9	202.0 ± 27.3	−0.44	0.66
HDL (mg/dL)	48.9 ± 11.8	51.9 ± 15.2	1.71	0.11
LDL (mg/dL)	127.5 ± 19.3	122.2 ± 20.4	−1.08	0.29
TG (mg/dL)	209.9 ± 125.6	186.1 ± 92.2	−0.85	0.41
HbA1C (%)	5.8 ± 1.0	5.5 ± 0.6	2.32	0.03 *
PFT				
FVC (liter)	4.2 ± 0.7	4.2 ± 0.9	−0.08	0.93
FEV1 (liter)	3.4 ± 0.6	3.4 ± 0.6	−0.95	0.35

PFT, pulmonary functional test; FVC, forced vital capacity; FEV1, forced expiratory volume in one second; HDL, high-density lipoprotein; LDL, low-density lipoprotein; TG, triglyceride; HbA1c, glycated hemoglobin. * *p* < 0.05.

**Table 4 healthcare-11-00106-t004:** Changes in self-reported measures (*n* = 14).

Assessment	Outcome Measure	Baseline	Post-12 Weeks	*t*-Value	*p*-Value
Self-reported	EQ-5D-3L (QoL)	16.3 ± 3.2	16.9 ± 2.1	0.59	0.56
	IPAQ (Physical activity)	4806 ± 3875.3	4855.1 ± 3849.9	0.11	0.91

QoL, quality of life; IPAQ, International Physical Activity Questionnaire.

## Data Availability

Not applicable.

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
