# Peer review of "Development of a Digital Healthcare Management System for Lower-Extremity Amputees: A Pilot Study"

_healthcare, 2022, doi:10.3390/healthcare11010106_

Round 1

Reviewer 1 Report

The purpose of this paper is to evaluate the effects of self-directed muscle strengthening training and cardiovascular exercise using a new ICT-based digital healthcare management service for lower limb amputation patients. In the discussion of the existing digital healthcare technology, there were not many discussions about specific medical fields. The topic of this thesis on how digital healthcare technology can specifically affect patients with lower limb amputations is judged to have great academic value. Although this paper has a small sample size and a short follow-up period after medical intervention, it is academically significant and requires future expansion.

Author Response

Thank you once again for the reviewers' valuable comments, and the readability and quality of this journal have improved thanks to the reviewers' valuable comments.

Reviewer 2 Report

Dear Authors:

First of all, congratulations on the work done. It is an interesting research but with certain limitations and formal errors that make me question the appropriateness of this manuscript in this Journal.

ABSTRACT:
The use of abbreviations in this section is discouraged. Please remove them.
This section should begin with a sentence that contextualises the object of the study, its importance and/or the appropriateness of the research.

INTRODUCTION:
This section should elaborate further on the object of study and the importance and impact of the results of this research.

METHODOLOGY:
This section should begin with the definition of the study methodology and its justification. This should be followed by a description of the characteristics of the study population and sampling. I advise authors, in general, to apply the Equator Network checklist best suited to their research design. This will allow authors to convey all the information necessary to ensure the reproducibility of the research and, moreover, to do so in the correct order.
Authors should include the power calculation of the sample size included.
The statistical analysis should be complemented by the calculation of the effect sizes of the applied statistical techniques.

RESULTS:
Tables and figures have abbreviations not described at the bottom of the tables and figures. Please correct the tables.
The zero as the last decimal place does not mean anything. Please remove zeros as last decimal place.

DISCUSSION:
This section would also benefit from the inclusion of more bibliographical references (as recommended for the Introduction section).
The limitations of the research should be more honestly formulated.

Kind regards

Author Response

Reviewer 2

Thank you for your valuable comments.

  1. ABSTRACT: The use of abbreviations in this section is discouraged. Please remove them.This section should begin with a sentence that contextualises the object of the study, its importance and/or the appropriateness of the research.

Answer: Thank you very much for your suggestion. We removed the abbreviation and rewritten it.

  1. INTRODUTION: This section should elaborate further on the object of study and the importance and impact of the results of this research.

Answer: Thanks for your suggestions, we have tried to elaborate on the experimental results in the discussion and conclusions. I would appreciate it if you could look at it with a wide tolerance.

  1. METHODOLOGY: This section should begin with the definition of the study methodology and its justification. This should be followed by a description of the characteristics of the study population and sampling. I advise authors, in general, to apply the Equator Network checklist best suited to their research design. This will allow authors to convey all the information necessary to ensure the reproducibility of the research and, moreover, to do so in the correct order.
    Authors should include the power calculation of the sample size included.
    The statistical analysis should be complemented by the calculation of the effect sizes of the applied statistical techniques.

Answer: We totally agree with reviewer’s comment. I think it will be easier for readers to understand the digital healthcare management system as a whole if they understand it first. I proceeded by referring to the Equator Network checklist suggested by the reviewer, but unfortunately it seems difficult to modify the order.

We added the following explanation for power calculation of the sample size.

“A minimum sample size of 15 subjects was calculated using a power calculation tool (G*Power 3.1.9.3 software; Heinrich-Heine University, Düsseldorf, Germany), with power, alpha, and effective size set at 0.80, 0.05, and 0.82, respectively.”

  1. RESULTS: Tables and figures have abbreviations not described at the bottom of the tables and figures. Please correct the tables.
    The zero as the last decimal place does not mean anything. Please remove zeros as last decimal place.

Answer: We totally agree with reviewer’s comment. The table has been modified.

“Table 2. K-Level, Lower limb extrenity prostehsis medicare functional classification levels”

  1. DISCUSSION: This section would also benefit from the inclusion of more bibliographical references (as recommended for the Introduction section).
    The limitations of the research should be more honestly formulated.

Answer: Thank you for your comment. We inserted a reference. The limitations of the study were added as follows.

“Moreover, there is no verification method for whether the patient's physical strengthening training and cardiovascular exercise performance rate was accurately measured.”

Thank you once again for the reviewers' valuable comments, and the readability and quality of this journal have improved thanks to the reviewers' valuable comments.

Reviewer 3 Report

I have reviewed a paper entitled  Development of information and communication technology-based digital healthcare management system for lower extremity amputees: A pilot study. Overall the paper is well-structured and overall well written. Bellow are some significant flawsthat I would like to highlight and should be revised, before the paper can be considered for publishing:

1. information and communication technology-based is redundant. I would assume a digital healthcare management system is quite obviously a computer/ICT-based solution.

2. Inconsistency in number of patients reported in abstract (n=14) and section 2.2. Participants (n=15) and Results (n=15) and Table 3. Changes in Clinical data (n=14)

3. The sample size might be sufficient, however, statistical justification is missing. Information under section 2.5 does not justify the number of subjects involved. I would really appreciate it if the authors could describe in detail how the sample size is justified.

4. there is a mistake in the URL: http;//clinicaltrials.gov (page 3, line 94). Please provide a proper reference to the protocol (i.e. at least a link), I could not find the study on https://clinicaltrials.gov/.

5. Background research is limited and superficial. Evaluation of current knowledge directly linked to the scientific question(s) to be answered by the study is missing. I would ask the authors to include the research question and discuss them in the discussion.

6. Study rationale is weakly expressed; i.e.  Motivation and justification of why this study was required are not clearly expressed.

7. How was the correctness of how patients execute the interventions measured/monitored? I would assume this should have a major effect on the outcomes. Is the study carried out at home (which seems to be the case) or under semi or full supervision? This information fits into the description of the intervention.

8. From experience, I would argue that calculating adherence based on video-play time is not sufficient and would assume some other mechanisms were deployed, at least during the follow-ups. Namely, one can do 0 exercises and just play the video and follow the app. Please integrate the ''additional'' measures taken into consideration into section 2.6.4 or mention this issue in the limitations. If I am mistaken, please justify this with a concrete reference where it clearly states that application activity is sufficient to measure adherence to interventions in an exercise app.

9. Also, I do not understand the formula for Adherence to cardiovascular exercise.  If I read correctly, adherence is measured by evaluating the time within which a patient's heart rate is between specific values. If I am correct, please simplify this in the paper.

  10. section 2.4 Measurement methods and items should list which are the methods and items. As it is written., this section is superficial, generic, and ultimately redundant, since sections 2.5 and 2.6 that follow actually outline the ''items'' in detail. It would also be of much interest to understand what are the primary and what secondary outcomes since this also reflects to my comment related to the sample size (remark 3)

11. Please outline whether English or Korean versions of the self-reported measures were used. I would assume Korean was used. If so please provide the proper references. If the version were not psychometrically validated, please outline the process under which the questionnaires were translated into Korean.   

12. ''Informed Consent Statement: Informed consent will be obtained from all subjects involved in the study'' I would assume, it was obtained.

Author Response

Reviewer 3

Thank you for your valuable comments.

  1. Information and communication technology-based is redundant. I would assume a digital healthcare management system is quite obviously a computer/ICT-based solution.

Answer: Thank you very much for your suggestion. I accepted the reviewer's suggestion and deleted the duplicated expression in the title and text. For example: Information and communication technology (ICT)-based digital healthcare management system -> digital healthcare management system.

  1. Inconsistency in number of patients reported in abstract (n=14) and section 2.2. Participants (n=15) and Results (n=15) and Table 3. Changes in Clinical data (n=14)

Answer: Thank you very much for your suggestion. Our study recruited 15 people, and one dropped out during the experiment. In the results, 14 people were indicated except for one dropout, and general characteristics of the 14 people were described in Table 3.

  1. The sample size might be sufficient, however, statistical justification is missing. Information under section 2.5 does not justify the number of subjects involved. I would really appreciate it if the authors could describe in detail how the sample size is justified.

Answer: We totally agree with reviewer’s comment. We added the following explanation for power calculation of the sample size.

  • Data and statistical analysis

“A minimum sample size of 15 subjects was calculated using a power calculation tool (G*Power 3.1.9.3 software; Heinrich-Heine University, Düsseldorf, Germany), with power, alpha, and effective size set at 0.80, 0.05, and 0.82, respectively.”

  1. There is a mistake in the URL: http;//clinicaltrials.gov (page 3, line 94). Please provide a proper reference to the protocol (i.e. at least a link), I could not find the study on https://clinicaltrials.gov/.

Answer:  Thank you very much for your suggestion. There was a typo. Edited.

 https://clinicaltrials.gov/.----> https://cris.nih.go.kr

  1. Background research is limited and superficial. Evaluation of current knowledge directly linked to the scientific question(s) to be answered by the study is missing. I would ask the authors to include the research question and discuss them in the discussion.

Answer: Thank you very much for your suggestion. There must be something missing in my research. There were many difficulties in researching the development of a digital healthcare system for amputee patients. There were limitations to the background research you pointed out. We hope that you will be kind enough to understand, and based on this research, we will conduct more in-depth research in the future.

  1. How was the correctness of how patients execute the interventions measured/monitored? I would assume this should have a major effect on the outcomes. Is the study carried out at home (which seems to be the case) or under semi or full supervision? This information fits into the description of the intervention.

Answer: Thanks for your suggestions, A very important question. We regularly provided education on accurate monitoring, intervention, and measurement methods to patients who participated in the study (education for all subjects before the experiment, individual education during the experiment). In addition, guardians were informed.

    In the study, self-directed physical strengthening training and cardiovascular exercise

were performed at home. I'll add info.

“In this study, the digital healthcare management system was used to perform home self-directed physical strengthening training and cardiovascular exercise cardiovascular exercise three times a week for 12 weeks.”

  1. From experience, I would argue that calculating adherence based on video-play time is not sufficient and would assume some other mechanisms were deployed, at least during the follow-ups. Namely, one can do 0 exercises and just play the video and follow the app. Please integrate the ''additional'' measures taken into consideration into section 2.6.4 or mention this issue in the limitations. If I am mistaken, please justify this with a concrete reference where it clearly states that application activity is sufficient to measure adherence to interventions in an exercise app.

Answer: We totally agree with reviewer’s comment. In the limitations of the study, the following explanation was added. “Moreover, there is no verification method for whether the patient's physical strengthening training and cardiovascular exercise performance rate was accurately measured.”

  1. Also, I do not understand the formula for Adherence to cardiovascular exercise.  If I read correctly, adherence is measured by evaluating the time within which a patient's heart rate is between specific values. If I am correct, please simplify this in the paper.

Answer: Thanks for your suggestions. Your understanding is correct. The description has been amended as follows:

“The calculation formula was as follows: 20min/day × 3 times/week = 60min, 30min/day × 3 times/week × 11 weeks = 990 min; physical activity time/1,050(60+990) min × 100 = adherence (%).”

  1. Section 2.4 Measurement methods and items should list which are the methods and items. As it is written., this section is superficial, generic, and ultimately redundant, since sections 2.5 and 2.6 that follow actually outline the ''items'' in detail. It would also be of much interest to understand what are the primary and what secondary outcomes since this also reflects to my comment related to the sample size (remark 3)

Answer: Thanks for your suggestions. It was judged that a description was necessary to help the reader's understanding.

  1. Please outline whether English or Korean versions of the self-reported measures were used. I would assume Korean was used. If so please provide the proper references. If the version were not psychometrically validated, please outline the process under which the questionnaires were translated into Korean.

Answer: Thank you for your comment. We inserted a reference.

“Kim, S. H., H. J. Kim, S.-i. Lee and M.-W. Jo. "Comparing the psychometric properties of the eq-5d-3l and eq-5d-5l in cancer patients in korea." Quality of life research 21 (2012): 1065-73.”

“Chun, M. Y. "Validity and reliability of korean version of international physical activity questionnaire short form in the elderly." Korean journal of family medicine 33 (2012): 144.”

  1. ''Informed Consent Statement: Informed consent will be obtained from all subjects involved in the study'' I would assume, itwas 

Answer: Thanks for your suggestions. Modified as below.

“Written informed consent was obtained from the participants.”

Thank you once again for the reviewers' valuable comments, and the readability and quality of this journal have improved thanks to the reviewers' valuable comments.

Round 2

Reviewer 2 Report

Dear Authors,

Congratulations on the corrections made. I now consider your manuscript suitable for publication.

Kind regards.

Author Response

(The authors gave the same response as above.)

Reviewer 3 Report

I would like to thank the authors for the review. The paper improved. Most of my comments were resolved. However, the main concern related to Related Works still exists.

I do appreciate the effort taken to design and evaluate the system. However, it is a research papers which should in principle require baseline understanding of related works on the topic of digital health systems and especially those related to amputation. The background research is still weak. I would strongly suggest the authors provide a paragraph or two in the introduction section on how researchers use digital health systems to solve issues in rehabilitation, if possible even in the context of amputation.

Author Response

Reviewer 3

Thank you for your valuable comments.

  1. I do appreciate the effort taken to design and evaluate the system. However, it is a research papers which should in principle require baseline understanding of related works on the topic of digital health systems and especially those related to amputation. The background research is still weak. I would strongly suggest the authors provide a paragraph or two in the introduction section on how researchers use digital health systems to solve issues in rehabilitation, if possible even in the context of amputation.

Answer: Thank you very much for your suggestion. We inserted a reference. The background of the study were added as follows.

“ The demand for healthcare services that enable health management, prevention, treatment, and post-management in daily life from existing treatment-oriented medical services has recently increased. Digital healthcare deals with personal health and medical information, devices, and platforms and is a comprehensive medical service that combines health-related services and medical IT. Personalized healthcare services can be provided based on health information such as lifestyle, physical examination, and medical use information obtained from portable and wearable devices owned by individuals or from cloud hospital information systems [6, 7]. Recently, a medical service model through a digital healthcare system has gained applications in the clinical field [8-11].”

Thank you once again for the reviewers' valuable comments, and the readability and quality of this journal have improved thanks to the reviewers' valuable comments.
